# Learning Disentangled Representations with Wasserstein Auto-encoders

**Paul Rubenstein,**[*] **Bernhard Schölkopf, Ilya Tolstikhin**
Empirical Inference
Max Planck Institute for Intelligent Systems, Tübingen
{paul.rubenstein,bs,ilya}@tuebingen.mpg.de

## Abstract

We apply Wasserstein auto-encoders (WAEs) to the problem of *disentangled representation learning*. We highlight the potential of WAEs with promising results on a benchmark disentanglement task.

## 1 Introduction

Wasserstein auto-encoders (WAEs) are a recently introduced auto-encoder architecture with justification stemming from the theory of Optimal Transport (Tolstikhin et al., 2018). Similarly to Variational auto-encoders (VAEs), WAEs describe a particular way to train probabilistic *latent variable models* (LVMs) $P_G$. LVMs act by first sampling a code (feature) vector $Z$ from a *prior distribution* $P_Z$ defined over the latent space $\mathcal{Z}$ and then mapping it to a random input point $X \in \mathcal{X}$ using a conditional distribution $P_G(X|Z)$ also known as *the decoder*.

Instead of minimizing the KL divergence between the LVM $P_G$ and the unknown data distribution $P_X$ as done by VAEs, WAEs aim at minimizing any optimal transport distance between them. Given any non-negative cost function $c(x, x')$ between two images, WAEs minimize the following objective with respect to parameters of the decoder $P_G(X|Z)$:

$$\min_{Q(Z|X)} \mathop{\mathbb{E}}_{P_X} \mathop{\mathbb{E}}_{Q(Z|X)} \big[ c\big(X, G(Z)\big) \big] + \lambda \mathcal{D}_Z(Q_Z, P_Z), \tag{1}$$

where the conditional distributions $Q(Z|X)$ are commonly known as *encoders*, $Q_Z(Z) := \int Q(Z|X)P_X(X)dX$ is *the aggregated posterior* distribution, $\mathcal{D}_Z$ is any divergence measure between two distributions over $\mathcal{Z}$, and $\lambda > 0$ is a regularization coefficient. In practice $Q(Z|X = x)$ and $G(z)$ are often parametrized with deep nets, in which case back propagation can be used with stochastic gradient descent techniques to optimize the objective. We will consider only *random* encoders $Q(Z|X = x)$ mapping inputs to a *distribution* over the latent space.

The objective (1) is similar to that of the VAE and has two terms. The first *reconstruction term* aligns the encoder-decoder pair so that the encoded images can be accurately reconstructed by the decoder as measured by the cost function $c$ (we will only use the *cross-entropy loss* throughout). The second regularization term is different from VAEs: it forces the aggregated posterior $Q_Z$ to match the prior distribution $P_Z$ rather than asking point-wise posteriors $Q(Z|X = x)$ to match $P_Z$ simultaneously for all data points $x$. This means that WAEs explicitly control the shape of the *entire* encoded dataset while VAEs constrain every input point separately.

In this work, we apply WAEs to the problem of *disentangled representation learning*, which is closely related to the more general problem of *manifold learning* for which auto-encoding architectures are often employed. The goal, though not precisely defined, is to learn representations of datasets such that individual coordinates in the feature space correspond to human-interpretable generative factors (also referred to as *factors of variation* in the literature). It is argued by Bengio et al. (2013) and Lake et al. (2017) that learning such representations is essential for significant progress in machine learning research.

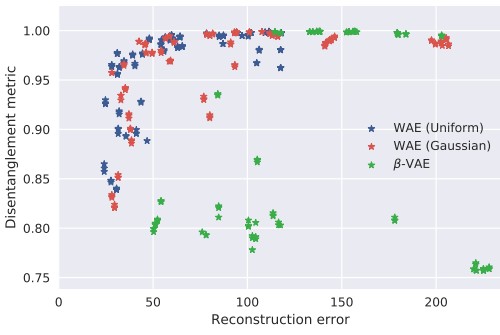
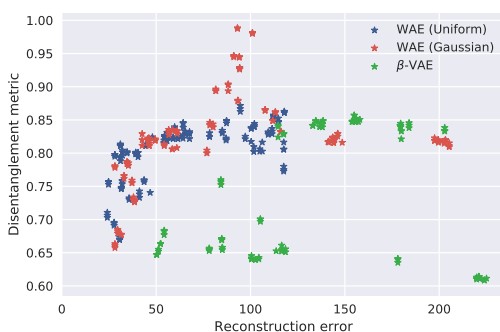

(a) 4-variable *dSprites* disentanglement task.       (b) 5 variable *dSprites* disentanglement task.

Figure 1: Disentanglement vs reconstruction error for $\beta$-VAEs with various values of $\beta$ and WAEs with various $L_1$ regularisation coefficients $\lambda_1$ **(up and left is better)**. Note that there is no direct way to compare different values of $\beta$ and $\lambda_1$, but in both cases increasing the value of the hyper-parameter is correlated with increasing reconstruction error. **WAEs are capable of achieving comparable or better disentanglement scores than the $\beta$-VAE while simultaneously achieving lower reconstruction errors.** In particular, WAE attains a maximum $98.8\%$ on the 5-variable disentanglement tast, compared to a maximum of $85.4\%$ for $\beta$-VAE)

## 2 EXPERIMENTS

Recently, Higgins et al. (2017) proposed the synthetic *dSprites* dataset and a metric to evaluate algorithms on their ability to learn disentangled representations. The dataset consists of 2-dimensional white shapes on a black background with 5 factors of variation: shape, size, rotation, $x$-position and $y$-position. Samples from this dataset can be seen in the first row of Figure 2.

The metric can be used to evaluate the "level of disentanglement" in the representation learned by a model when the ground truth generative factors are known for each image, such as for the *dSprites* dataset. We provide here an intuition of what the metric does; see Higgins et al. (2017) for full details. Given a trained feature map $\varphi \colon \mathcal{X} \to \mathcal{Z}$ from the image space to the latent space, we ask the following question. Suppose we are given two images $x_1$ and $x_2$ which have exactly one latent factor whose value is the same—say they are both the same shape, but different in size, position and rotation. By looking at the *absolute values of the difference in feature vectors* $|\varphi(x_1) - \varphi(x_2)| \in \mathbb{R}^{d_{\mathcal{Z}}}$, is it possible to identify that it is the *shape* that they share in common, and not any other factor? The idea is that if a disentangled representation has indeed been learned, then for each latent factor there should be some feature coordinate $\varphi_i$ corresponding to it. The value of $|\varphi_i(x_1) - \varphi_i(x_2)|$ should then be close to zero for the latent factor that is shared, while other coordinates should on average be larger.

In the same paper, the authors introduce the $\beta$-VAE, which is currently considered to be the state-of-the-art in disentangled learning algorithms. The $\beta$-VAE is a modification of the original VAE in which the KL regularisation term is multiplied by a scalar hyper-parameter $\beta$. The authors show that by tuning $\beta$, they are able to explore a trade-off between entangled representations with low reconstruction error and disentangled representations with high reconstruction error.

We replicated the main experiment performed by Higgins et al. (2017) on the *dSprites* dataset, which we describe in brief here. For further details, we refer the reader to Section 4.2 and Appendix A.4 of their paper. We used a fixed fully connected architecture with the Bernoulli reconstruction loss for all experiments, with a latent space dimension of 16. We trained 10 $\beta$-VAEs for $\beta \in \{1, 3, 10, 20, 30, 40, 50, 75, 100\}$. For each of the 10 replicates of each value of $\beta$, we calculated the disentanglement metric 3 times. From the resulting list of 30 numbers, we discarded the bottom 50%. For each experiment, we also record the test reconstruction error on a held out part of the dataset. At the end of this procedure we had 15 pairs of numbers (test reconstruction error, disentanglement) for each of the 9 choices of $\beta$.

---

*Also affiliated with: Machine Learning Group, Engineering Department, University of Cambridge

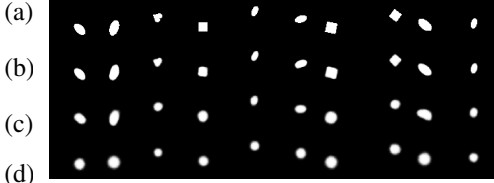

(a)

(b)

(c)

(d)

Figure 2: **Row (a)**: Samples from the *dSprites* dataset; the remaining rows show reconstructions of these images by: **Row (b)**: the Gaussian WAE with the best reconstruction error amongst those scoring $> 98\%$ on the 4-variable disentanglement metric; **Row (c)**: the Gaussian WAE with the best score on the 5-variable disentanglement metric; **Row (d)**: the $\beta$-VAE with the best reconstruction error amongst those scoring $> 98\%$ on the 4-variable disentanglement metric. This visually confirms what is shown in Figure 1, namely that WAEs can disentangle better than $\beta$-VAEs while preserving better reconstructions.

We repeated the same process with two types of random-encoder WAEs sharing the same architectures as the $\beta$-VAE for the encoder and decoder. The first type had Gaussian priors and Gaussian encoders. The second type had a uniform prior on $[-1, 1]^{d_{\mathcal{Z}}}$ and uniform encoder[1] mapping to axis-aligned boxes in $\mathcal{Z}$. In both cases, the means of the encoders were constrained to be in the range $(-1, 1)$ on each dimension by *tanh* activation functions. We additionally added $L_1$ regularisation to the log-variances - doing so encourages the encoders to remain stochastic by preventing the log-variances becoming too negative. More precisely, we added the following term to the objective function to be minimised:

$$\frac{\lambda_p}{N} \sum_{n=1}^{N} \sum_{i=1}^{d_{\mathcal{Z}}} \left| \log \left( \sigma_i^2(x_n) \right) \right|^p \tag{2}$$

where $i$ indexes the dimensions of the latent space $\mathcal{Z}$, $n$ indexes the inputs in a mini-batch and $\lambda_p \geq 0$ is a new regularization coefficient. We trained such WAEs with $L_1$ regularisation coefficients $\lambda_1 \in \{0, 0.1, 0.5, 1, 2, 3, 5, 8, 12\}$.

Higgins et al. (2017) report their results for disentangling on only 4 of the possible 5 variables.[2] We additionally calculated the disentangling metric on the more challenging task of distinguishing between *all 5* of the latent variables. The results of our experimentation are displayed in Figure 1. We were able to replicate their results showing that the $\beta$-VAE is capable of achieving essentially $100\%$ on the 4-variable disentanglement task (Figure 1a), and that good disentanglement of $\beta$-VAE comes at the expense of poorer reconstruction. On the 4-variable disentanglement task, we found that WAEs were able to attain similar levels of disentanglement while retaining significantly better reconstruction errors. On the 5-variable task (Figure 1b), WAEs significantly outperformed $\beta$-VAEs simultaneously in terms of disentanglement and reconstruction.

Amongst all of the $\beta$-VAEs we trained attaining a 4-variable disentanglement score of $> 98\%$, the lowest training reconstruction error was $114.8 \pm 5.2$. The corresponding error for the WAEs was $40.8 \pm 1.5$. The WAE with the best 5-variable disentanglement scored an average of **98.8**$\%$ across the 3 independent disentanglement calculations for this experiment with a test reconstruction of $94.0 \pm 5.8$. The $\beta$-VAE performing best on the 5-variable disentanglement task scored an average of **85.4**$\%$ disentanglement with a test reconstruction of $156.1 \pm 5.7$. In summary, WAEs are able to outperform $\beta$-VAEs simultaneously in terms of disentanglement metric and reconstruction error. Sample reconstructions from each of the aforementioned experiments are displayed in Figure 2.

## 3 CONCLUSION

Preliminary experimentation suggests that Wasserstein auto-encoders may be a useful tool for disentangled representation learning. Future directions for research include investigating the role of more complex priors that can be composed together to model structured data, and whether optimal values of the $\lambda_1$ hyper-parameter can be found adaptively by the learning machine itself.

---

[1]Here the *log-side-lengths* were parametrised by the encoder, not the log-variances.

[2]Although there are 5 factors of variation in the *dSprites* dataset, the number $99.23 \pm 0.1\%$ they reported in the Figure 6 of the main section of the paper refers to the ability of the $\beta$-VAE to provide a feature map with which a classifier can predict whether $x$-position, $y$-position, scale and rotation are shared between pairs of images, while ignoring shape. This is stated in Appendix A.4 of their paper.

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
