# OpenReview forum: "Learning Disentangled Representations with Wasserstein Auto-Encoders"
_ICLR.cc/2018/Workshop — Accept_

### Official Review · AnonReviewer2 · 2018-03-10

**Rating:** 6
**Confidence:** 3

**Review:**


The work shows some preliminary results that suggest that Wasserstein autoencoders could be used for finding disentangled representations. The method performs comparable (or better) than the beta-VAE on a toy but illustrative example.

The experimental evaluation is limited and comparisons only include beta-VAE, which is of course a key baseline. The paper would be stronger if other real datasets were evaluated (say some simple image datasets, like CelebA or 3D faces, as in beta-VAE), and other baselines were included.

How do results vary with lambda? Results in the 5-variable sprites seem to be very good only for a small set of the examples.

How much does the variance regularization term term contribute to the results?

Other minor comments:

- The affiliation footnote is on page 2 rather than page 1.
- It would be helpful if the plots in Figure 1 where on the same scale.

---

> ### Public Comment · ~Paul_K_Rubenstein1 · 2018-03-21
> **Reply**
>
> Many thanks for your review.
>
> The problem with using real datasets such as CelebA for evaluating levels of disentanglement in the learned representation is that since there is not a one-one correspondence between labels and images, evaluation of disentanglement typically has to be done by plotting walks along different axes and visually inspecting the output. While there is nothing in principle wrong with this, it makes objective comparison between different algorithms difficult. Though we thank you for pointing out that including such plots would improve the paper.
>
> How do results vary with lambda? We did not thoroughly investigate the role of lambda (the weighting of the D(P_Z, Q_Z) term) on disentanglement. We simply fixed a value for this for which we observed good performance of WAEs on benchmark tasks not related to disentanglement.
>
> We agree that results in the 5-variable sprites seem to be good only for a small set of the parameters we searched over. There is still progress to be made, even in this simple problem.
>
> How much does the variance regularization term contribute to the results? This is the key to our results. We show that tuning this parameters makes the difference between very poor and very good performance.

---

### Official Review · AnonReviewer1 · 2018-03-17
**Could the authors discuss the relationship between WAEs and the InfoVAE?**

**Rating:** 7
**Confidence:** 4

**Review:**

The authors evaluate disentangled representations of the dSprites data learned with Wasserstein auto-encoders to those learned with a β-VAE. They find that that WAEs yield lower reconstruction errors when at similar levels of accuracy of the disentanglement metric proposed by Higgins et al.

This seems like a completely reasonable workshop submission to me. My only point of criticism would be that the authors could do a better job of comparing to related (and in some cases very recent) related work. There have been a relatively large number of recent proposals for modifications of the VAE objective aid learning of disentangled representations. Many of these objectives incorporate terms that depend on the aggregated posterior. This includes objectives that minimize the total correlation between latent variables such as proposed by Kim and Mnih, as well as (more recently) Chen et al. and Gao et al.

Perhaps the most closely related example is the InfoVAE by Zhao et al, which if I understand correctly is equivalent to the WAE when c is the cross entropy and D_Z is the KL divergence (although admittedly I was not immediately able to figure out from the Wasserstein VAE paper whether there is some optimal transport stuff that factors into the WAE objective in a way that I don't understand). Assuming this method is not equivalent to the InfoVAE, then it would be interesting to see a direct comparison between these two methods. In other words, do we get better disentangled relative to the β-VAE because we regularize using a D_Z(Q_Z, P_Z) that compares the aggregate posterior to the prior (which is also the case in the InfoVAE) or is it the case that WAEs also outperform the InfoVAE?

On a related note, I was not able to find what the divergence measure D_Z(Q_Z, P_Z) the authors use. Is this a GAN-style objective, the MMD, or simply the KL divergence? Note in this context that Kim & Mnih also use a GAN-style discriminator to minimize the total correlation.

Minor

- What is the difference between P_G(X | Z) and G(Z) here?

References

1. Hoffman, M. D. & Johnson, M. J. Elbo surgery: yet another way to carve up the variational evidence lower bound. in Workshop in Advances in Approximate Bayesian Inference, NIPS (2016).

2. Zhao, S., Song, J. & Ermon, S. InfoVAE: Information Maximizing Variational Autoencoders. arXiv:1706.02262 [cs, stat] (2017).

3. Kim, H. & Mnih, A. Disentangling by factorising. arXiv preprint arXiv:1802.05983 (2018).

4. Chen, T. Q., Li, X., Grosse, R. & Duvenaud, D. Isolating Sources of Disentanglement in Variational Autoencoders. arXiv:1802.04942 [cs, stat] (2018).

5. Gao, S., Brekelmans, R., Steeg, G. V. & Galstyan, A. Auto-Encoding Total Correlation Explanation. arXiv:1802.05822 [cs, stat] (2018).

---

> ### Public Comment · ~Paul_K_Rubenstein1 · 2018-03-21
> **Thanks for the related works; InfoVAE is discussed in WAE paper**
>
> Many thanks for your review.
>
> We were aware of some (but not all) of the related works you mentioned, so many thanks for pointing them out. We look forward to these works being accepted for publication so that in future work we can thoroughly compare them with our work.
>
> InfoVAE is indeed essentially the same as WAE, though derived in a different way with a different motivation. The relationship is discussed in the latest arxiv version of the Wasserstein Auto-Encoders paper: https://arxiv.org/pdf/1711.01558.pdf
>
> The divergence measure we used was the MMD.
>
> P_G(X|Z) and G(z): There was a typo, we should have written P_G(X|X=z) rather than G(z). Thank you for noticing this.

---

### Decision · Program_Chairs · 2018-03-20
**ICLR 2018 Workshop Acceptance Decision**

**Decision:**

Accept

**Comment:**

Congratulations, your paper was accepted to the ICLR workshop.